# Learning to Represent Whole Slide Images by Selecting Cell Graphs of Patches

**Yinan Zhang** [*]                                                          ZHANGYINAN9@JD.COM
**Beril Besbinar** [*]                                                      BERIL.BESBINAR@EPFL.CH
**Pascal Frossard**                                                    PASCAL.FROSSARD@EPFL.CH
*Signal Processing Laboratory (LTS4), École Polytechnique Fédérale de Lausanne (EPFL)*

## Abstract

Advances in multiplex biomarker imaging systems have enabled the study of complex spatial biology within the tumor microenvironment. However, the high-resolution multiplexed images are often only available for a subset of regions of interest (RoIs), clinical data is not easily accessible and the datasets are generally too small to apply off-the-shelf deep learning methods commonly used in histopathology. In this paper, we focus on datasets with few images and without labels and aim to learn representations for slides. We choose a task of patient identification that leads our new model to select RoIs with discriminative properties and infer patient-specific features that can be used later for any task via transfer learning. The experimental results on the synthetic data generated by taking the tumor microenvironment into account indicate that the proposed method is a promising step towards computer-aided analysis in unlabeled datasets of high-resolution images.

## 1. Introduction

The immune microenvironment is believed to carry valuable information for cancer prognosis and personalized medicine (Whiteside, 2008). Recent progress in cell segmentation and classification or automated image analysis software has facilitated the use of cell-graphs (Yener, 2016) with network analysis tools for the investigation of structure-function relationship in tumour microenvironment. In particular, the multiplex biomarker imaging systems deliver precise outcomes for such analysis, such as the cell phenotype for a diverse set of markers or the morphological properties of cells. However, as the technology is relatively recent, access to big datasets of such slides with accompanying clinical data is often limited. Moreover, in practice, only some regions of interest (RoIs) are available at high resolution.

Here, we consider a scenario where few slides are available with no associated labels, and every slide is described by cell level analysis of RoIs extracted from different parts of the slide. We use the patient identification problem as the task to aid learning representations for whole slides using a chosen subset of RoIs that help to distinguish a patient from the others by considering the spatial organization of their different cells. The slide-specific representations learned in this self-supervised way can be easily used for any supervised task with transfer learning, almost imitating a bottom-up clustering approach. For this purpose, we first describe each RoI as cell-graphs, where nuclei centers form the nodes, which are later processed using graph neural networks (GNNs). The resultant embeddings are then aggregated using a permutation invariant function to obtain the whole-slide representation, which is used for pretraining with the imposed classification task. We finally learn how to select a subset of ROIs using pre-trained ROI embeddings and stochastic layers with discrete random variables, which, in return, offers interpretability for the final decision.

---

[*] Contributed equally

1. The first author performed the work while at LTS4, EPFL

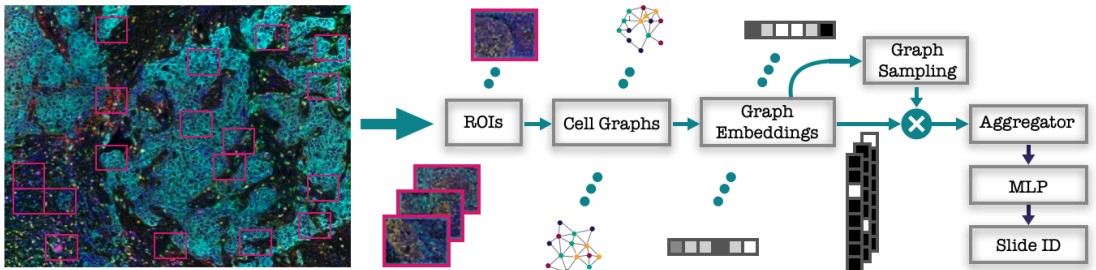

Figure 1: Multi-graph classification with learned graph selection.[2]

## 2. Proposed Method

The end-to-end differentiable proposed model is illustrated in Figure 1[2]. We assume that each slide has $n$ different ROIs, with cell nuclei centers and phenotypes are available. The cell graphs are constructed by thresholding the Euclidean distance between each pair of cells and the phenotype for each cell is encoded as one-hot vector as the initial cell features.

**Graph Embeddings and Pre-training.** The hierarchical graph embedding module is composed of successive GrapSAGE (Hamilton et al., 2017) and DiffPool (Ying et al., 2018) layers. GraphSAGE layers learn node embeddings by taking neighboring cells and their phenotypes into account. DiffPool layers leads to hierarchical abstraction of representations by clustering input nodes and jointly learning cluster embeddings. The uppermost DiffPool layer output is considered to be the ROI embedding, and it is pre-trained with the so-called patient identification task, where the slide-level embedding is obtained via a permutation-invariant aggreator, such as element-wise summation or maximum operator across all $n$ embeddings belonging to the same slide. Using fully connected layers (MLPs) followed by a softmax layer on top, we pre-train the embeddings with the well-known cross-entropy loss.

**Patch Selection.** All $n$ embeddings for each slide are used as input to another set of MLP layers, with the uppermost layer having $n$ neurons, to learn a discrete stochastic latent variable using the Gumbel-Softmax trick (Jang et al., 2016). By sampling the learned variable, $\mathbf{z}$, $K$ times and taking the dot-product with RoI embeddings, we obtain $K$ RoI embeddings, which are then used for the second phase of training with MLPs followed by a permutation invariant aggreagator and a softmax layer. The graph embedding layers continue to be updated with a smaller learning rate while learning the selection mechanism.

## 3. Experiments

**Dataset** We generate a synthetic dataset to imitate the the tumour microenvironment (Balkwill et al., 2012). We create an illustrative dataset of 10 patients, each represented by 15 cell graphs. The total number of nodes for each graph is fixed to 1000. We assume six different types of cells, two of which represent *tumour* and *stroma* cells, while the rest resemble lymphocytes. The majority of graphs are composed of uninfiltrated tumour-stroma border, while the rest is patient-specific combination of discriminative graphs. To represent 10 patients, we create five discriminative cell graphs with different combinations of lymphocyte types and percentages with different spatial organization. The specifics about the synthetic dataset can be seen in Table 1.

---

2. Image taken from https://www.leicabiosystems.com/

| | Ratio of Different Types of Cells | | | | | | Discriminative Patches | | | | | | | | | |
|---|---|---|---|---|---|---|---|---|---|---|---|---|---|---|---|---|
| | Tumour | Stroma | T-Cells | B-Cells | NK-Cells | Macrophages | Patient 1 | Patient 2 | Patient 3 | Patient 4 | Patient 5 | Patient 6 | Patient 7 | Patient 8 | Patient 9 | Patient 10 |
| Patch 1 | 0.40 | 0.40 | 0.10 | 0.10 | 0.00 | 0.00 | X | X | X | X | | | | | | |
| Patch 2 | 0.30 | 0.30 | 0.40 | 0.00 | 0.00 | 0.00 | X | | | | | X | X | X | | |
| Patch 3 | 0.30 | 0.30 | 0.20 | 0.20 | 0.00 | 0.00 | | X | | X | | | | X | X | |
| Patch 4 | 0.30 | 0.30 | 0.15 | 0.15 | 0.00 | 0.00 | | | X | | X | | X | | | X |
| Patch 5 | 0.30 | 0.30 | 0.10 | 0.10 | 0.10 | 0.10 | | | | X | | | X | | X | X |

Table 1: Characteristics of Synthetic Dataset.

Figure 2: Identification accuracy

**Results.** We use equal splits for training/validation/testing and the location of discriminative graphs are shuffled at each training step and test realization. We experimented with different aggregators, different architectures and *soft* and *hard* selection of patches. We here report the best performance out of all different settings, which is obtained with element-wise mean aggregator and soft selection mechanism. In Figure 3, the confusion matrix corresponding to the average of 100 realizations of sampling two graphs, i.e., two patches, per patient is presented. The corresponding percentage of selecting both ground truth discriminative patches correctly, selecting one out of two patches correctly and not being able to select any right patches correctly are 49.1%, 50.8% and 0.1%, respectively.

## 4. Conclusion

In this paper, we proposed a novel pipeline that uses cell graphs to represent high resolution images of RoIs, learns to select a subset of them and computes a hierarchical slide level representation from an unlabeled dataset on an representative patient identification task. The initial results on the synthetic dataset motivates the use of proposed method as a pre-training strategy or for supervised methods with minor modifications.

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
