# OpenReview forum: "Learning to Represent Whole Slide Images by Selecting Cell Graphs of Patches"
_MIDL.io/2021/Conference/Short — MIDL 2021 Poster_

### Official Review · Reviewer_mCZU · 2021-04-30

**Confidence:** 3
**Final Rating:** 2

**Summary:**

The authors present a pipeline consisting of neural networks to learn embeddings of high-resolution images (slides). For each slide, n specific ROIs are determined and represented as graphs. A graph neural network (GNN) transforms these graphs into embedding vectors that are aggregated to represent the whole slide. An additional MLP uses this representation to provide a class label; in this case, patient id classification is used to train the proposed model. Another MLP learns a discrete latent variable that is sampled K times to select a subset of the initial n embeddings. In a second training phase, the patient identification network is fine-tuned using the K sampled embeddings.

**Strengths:**

The authors present an approach for how GNNs can be used to represent large high-resolution images efficiently. The networks are trained to classify patient ids, a  task where no manual labeling process is involved.

The resulting embeddings can, in principle, be used to solve other supervised tasks where labels are costly. Selecting a descriptive subset of the presented ROIs may indicate where the network's attention is focused. In the experiments, almost every time, at least one of the known discriminative patches was selected. Although attention mechanisms are widespread in the related literature, the proposed sampling mechanism is novel in this context.

**Weaknesses:**

It is hard to follow the author's description of the experiments. Furthermore, the paper lacks a discussion that explains why the results are promising. The contribution and novelty of this work are not clearly stated; other related works are scarcely mentioned.

The proposed experiments are very preliminary. Although the approach is interesting, showing that it works on a small set of randomly generated graphs merely indicates that this semi-supervised approach may be used on a real dataset. The author's claims that their method i.) can easily be used for transfer learning of any supervised task and ii.) offers interpretability, are not backed up by the experimental results:



1.)  No experiments on transfer learning were conducted. Since the data are only simulated, why not also simulate a classification task to prove that the learned embeddings are effective?

*2.)* There is no comparison to a baseline. A simple histogram analysis may generalize on the proposed dataset since, based on Figure 1, the class-separating feature seems to be the count of cells-types for discriminative patches. This hints that the simulated task may be easy to learn. From this perspective, the good classification results are not surprising.

*3.)*  The patch selection results are not entirely convincing: in only 49.1% of the cases, both discriminative patches are selected correctly.

*4.)* A drawback of the sampling approach is that a value for K needs to be specified. Does increasing or decreasing K change the performance of the proposed method?



**Deanonymize Review:**

no

**Detailed Comments:**

- Why is cell segmentation specifically referenced, although no segmentation is performed in this work? Instead, a more related paper could have been referenced.
- A comparison of the approach with and without (first training stage vs. second training stage) patch selection could show whether the gained interpretability leads to performance losses.



**Justification Of The Rating:**

The approach is only moderately novel; many papers consider GNNs and cell graphs for example [1](https://link.springer.com/chapter/10.1007/978-3-030-00934-2_20), [2](https://ieeexplore.ieee.org/abstract/document/9288001?casa_token=FMkuALXxgzoAAAAA:7IDOmIO9RVHIkM2Tbf9eMGYsIkljQArGrM92gLBv2i5n5HtmkzJChGs9S2wkHaoiQtzv8eRfdDDz), [3](https://openaccess.thecvf.com/content_ICCVW_2019/html/VRMI/Zhou_CGC-Net_Cell_Graph_Convolutional_Network_for_Grading_of_Colorectal_Cancer_ICCVW_2019_paper.html). Furthermore, using the patient id to learn representations can be seen as a basic method of [self-supervised learning](https://ieeexplore.ieee.org/abstract/document/7312476). Nevertheless, the patch selection method appears to be novel.

However, this novelty does not outperform the other drawbacks of the work: the paper is not easy to understand,  only a few related works are mentioned, and when considering transfer learning or explainable AI, the results are not convincing. Moreover, they are insufficiently discussed.

**Paper Type:**

methodological development

**Special Issue:**

no

---

### Official Review · Reviewer_PS4z · 2021-04-30

**Confidence:** 4
**Final Rating:** 3

**Summary:**

This paper identifies a novel application of machine learning in high-resolution multiplexed images. The authors proposed to use the patient-identification task and Graph Neural Network (GNN) to learning representations for the slide from the spatial distribution of various types of nuclei in the cancer tissue as revealed by the multiplex images.

**Strengths:**

As far as I know, the authors identified a new application of machine learning in multiplex biomarker imaging. The spatial distribution of various types of nuclei in cancer tissue revealed by this type of image contains valuable information of scientific and clinical interest. And the authors proposed an interesting approach to address some of the challenges in this application (e.g., only a few ROIs are available).

**Weaknesses:**

1. The main weakness is that the method is only evaluated on a synthetic dataset. The process of generating the synthetic dataset is not very clear, e.g., how exactly are the different types of spatial organizations of nuclei generated?

2. The ablations studies are only mentioned but not reported in the results section. Thus the advantage of the proposed method is not very convincing. For example, is the ROI selection mechanism necessary? Can good patient identification performance be achieved without ROI selection? How sensitive is it with respect to how the synthetic data is generated?

**Deanonymize Review:**

no

**Detailed Comments:**

1. It would be great if the author can cite a paper related to the multiplex image method, which could be helpful for people who are not familiar with this area;

2. Figure 2 is referred to as Figure 3 in the text.

**Justification Of The Rating:**

Despite the shorting comings, I think the authors identified an interesting problem in pathology research and proposed an intriguing self-supervised learning task. This could be a thought-provoking paper at the conference.

**Paper Type:**

methodological development

**Special Issue:**

no

---

### Meta-Review · Program_Chairs · 2021-05-11

**Recommendation:** Accept (Poster)
**Confidence:** 5

**Metareview:**

Two reviews have different opinions about this work. The main weakness of this paper is that it did not report the ablation studies, so that the advantage of the proposed method is unclear. However, the authors propose a new application in multiplex biomarker imaging, which can make contribution to the community. I maintain the acceptance for this conference.

---

### Decision · Program_Chairs · 2021-05-11

Accept (Poster)